A new species of Allodaposuchus (Eusuchia, Crocodylia) from the Maastrichtian (Late Cretaceous) of Spain: phylogenetic and paleobiological implications

Blanco Alejandro 1
Fortuny Josep 1
Vicente Alba 2
Luján Àngel H. 1
García-Marçà Jordi Alexis 1
Sellés Albert G. 1 albert.garcia@icp.cat
1 Institut Català de Paleontologia Miquel Crusafont, Universitat Autònoma de Barcelona , Sabadell, Catalonia , Spain
2 Departament d’Estratigrafia, Paleontologia i Geociències marines, Facultat de Geologia, Universitat de Barcelona, Carrer de Martí i Franquès s/n , Barcelona, Catalonia , Spain
Anquetin Jérémy
Electronic publication date: 2015 Aug 13
Publication date: 2015
Volume: 3
Electronic Location ID: e1171
Received 2015 Mar 20; Accepted 2015 Jul 20
Copyright: © 2015 Blanco et al.
Copyright year: 2015
Copyright holder: Blanco et al.
License: This is an open access article distributed under the terms of the Creative Commons Attribution License, which permits unrestricted use, distribution, reproduction and adaptation in any medium and for any purpose provided that it is properly attributed. For attribution, the original author(s), title, publication source (PeerJ) and either DOI or URL of the article must be cited.
License URL: https://creativecommons.org/licenses/by/4.0/

Keywords: Crocodylia, Endocranial morphology, Locomotion, Allodaposuchus, Paleoecology, Late Cretaceous

Funding: Spanish Ministry of Economy and Competitiveness (MINECO) CGL2011-30069-C02-01/BTE CGL2011-27869 Generalitat de Catalunya (Catalan Government) FI-DGR 2013FI_B 01059 FI-DGR 2012FI_B 01221 Spanish Ministry of Economy and Competitiveness BES-2012-057837 This paper is a contribution to the projects CGL2011-30069-C02-01/BTE and CGL2011-27869 subsidized by the Spanish Ministry of Economy and Competitiveness (MINECO). The research of AB and ÀHL is supported by predoctoral grants (FI-DGR 2013FI_B 01059 and FI-DGR 2012FI_B 01221, respectively) awarded by the Generalitat de Catalunya (Catalan Government), whereas AV is granted by the Spanish Ministry of Economy and Competitiveness (BES-2012-057837). The funders had no role in study design, data collection and analysis, decision to publish, or preparation of the manuscript.

==============================
Background. The Late Cretaceous is a keystone period to understand the origin and early radiation of Crocodylia, the group containing all extant lineages of crocodilians. Among the taxa described from the latest Cretaceous of Europe, the genus Allodaposuchus is one of the most common but also one of the most controversial. However, because of its fragmentary record, several issues regarding its phylogenetic emplacement and its ecology remain unsolved or unknown. The discovery of a single specimen attributed to Allodaposuchus, represented by both cranial and postcranial remains, from the Casa Fabà site (Tremp Basin, NE Spain) in the lower red unit of the Tremp Fm. (early Maastrichtian, Late Cretaceous) offers a unique opportunity to deepen in the phylogenetic relationships of the group and its ecological features.

Methods. The specimen is described in detail, and CT scan of the skull is performed in order to study the endocranial morphology as well as paratympanic sinuses configuration. In addition, myological and phylogenetic analyses are also carried out on the specimen for to shed light in ecological and phylogenetic issues, respectively.

Results. The specimen described herein represents a new species, Allodaposuchus hulki sp. nov., closely related to the Romanian A. precedens. The CT scan of the skull revealed an unexpected paratympanic sinuses configuration. Allosaposuchus hulki exhibits an “anterodorsal tympanic sinus” not observed in any other extant or extinct crocodilian. The caudal tympanic recesses are extremely enlarged, and the expanded quadratic sinus seems to be connected to the middle-ear channel. Phylogenetic analyses confirm the emplacement of the informal taxonomic group ‘Allodaposuchia’ at the base of Crocodylia, being considered the sister group of Borealosuchus and Planocraniidae.

Discussion. Although this is a preliminary hypothesis, the unique paratympanic configuration displayed by A. hulki suggests that it could possess a high-specialized auditory system. Further, the large cranial cavities could help to reduce the weight of the cranium. Concerning the postcranial skeleton, Allodaposuchus hulki shows massive and robust vertebrae and forelimb bones, suggesting it could have a bulky body. The myological study performed on the anterior limb elements supports this interpretation. In addition, several bone and muscular features seem to point at a semi-erected position of the forelimbs during terrestrial locomotion. Taking all the above results into consideration, it seems plausible to suggest that A. hulki could conduct large incursions out of the water and have a semi-terrestrial lifestyle.

Introduction

The Late Cretaceous is a crucial period for understanding the rise and radiation of Crocodylia. At this time, the three main lineages of modern crocodilians made their appearance, and started their dominance upon other crocodilian faunas (Puértolas, Canudo & Cruzado-Caballero, 2011). Thus, any new find regarding the Eusuchia record is worthwhile because it provides new information to the puzzling origin of modern crocodilians.

In this way, the fossil record of Late Cretaceous crocodylomorphs from Europe offers an exceptional opportunity to approach such questions, because it contains both basal Eusuchians and members of all groups involved in the radiation of Crocodylia. In the uppermost Cretaceous strata of SW Europe, basal Eusuchia are represented by the hylaeochampsids Iharkutosuchus, Ősi, Clarck & Weishampel, 2007, Acynodon Buscalioni, Ortega & Vasse, 1997, and Musturzabalsuchus Buscalioni, Ortega & Vasse, 1997 (see also Narváez et al., 2014). In turn, the clade Alligatoroidea is represented by the genera Massaliasuchus Martin & Buffetaut, 2008, whereas Thoracosaurus Leidy, 1852 is included within Gavialoidea (see Laurent, Buffetaut & Le Loeuff, 2000), and Arenysuchus Puértolas, Canudo & Cruzado-Caballero, 2011 is considered a basal crocodyloid.

AllodaposuchusNopcsa, 1928 was one of the most common taxa during the latest Cretaceous of Europe, but is also considered one of the most controversial. Mainly represented by fragmentary skull remains, this genus currently comprises three different species (Allodaposuchus precedens Nopcsa, 1928, Allodaposuchus subjuniperus Puértolas-Pascual, Canudo & Moreno-Azanza, 2014, and Allodaposuchus palustris Blanco et al., 2014) reported from Spain (Buscalioni et al., 2001; Blanco et al., 2014; Puértolas-Pascual, Canudo & Moreno-Azanza, 2014), France (Martin, 2010), and Romania (Nopcsa, 1928; Delfino et al., 2008). From a phylogenetic point of view, Allodaposuchus has been considered for a long time to be a sister taxon of the family Hylaeochampsidae (see Buscalioni et al., 2001; Delfino et al., 2008; Puértolas-Pascual, Canudo & Moreno-Azanza, 2014), but has also been included within Alligatoroidea (Martin, 2010) or more recently treated as a basal crocodylian (Blanco et al., 2014).

In addition, until the discovery of A. palustris (Blanco et al., 2014), the features of the postcranial elements of this genus were never studied in detail. The recovery of new cranial and abundant postcranial material ascribed to Allodaposuchus from Casa Fabà site (Tremp Basin Southern Pyrenees; Fig. 1), not only sheds light on the anatomical characteristics of this genus, but also provides new clues for its systematic placement and paleobiological traits.

Figure 1 Geographical and geological location of the Casa Fabà site.

(A) Geological map of the Tremp Basin (modified from López-Martínez & Vicens, 2012); (B) stratigraphical section performed near the Casa Fabà site (modified from Riera et al., 2009); (C) mapping of the crocodilian bones at the Casa Fabà locality.

Geological Setting

The Casa Fabà locality is one of the dozens of Late Cretaceous continental fossil sites located within the Tremp Basin (Southern Pyrenees, Catalonia; Riera et al., 2009). Discovered by Ana María Bravo and Rodrigo Gaete in 2001, the Casa Fabà site is located about 500 m east of the village of Orcau (Pallars Jussà, Catalonia, Spain), in a ravine area known as Les Olives (Fig. 1A). At the end of the Cretaceous, the southern Pyrenean region (NE Iberian Peninsula) consisted of an elongated E-W foreland trough connected to the Atlantic Ocean. In this basin, sedimentation occurred in marine settings up to the Campanian-Maastrichtian boundary. Since then, the sedimentary environment gradually evolved to more continental conditions. As a result of an uplift of successive thrust-sheets involved in the formation of the Pyrenean range (Muñoz, Martínez & Vergés, 1986; Puigdefàbregas & Souquet, 1986), four synclines can now be distinguished from the east to the west: the Vallcebre syncline, the Coll de Nargó syncline and the Tremp and Àger synclines.

In the Tremp syncline, the uppermost Cretaceous non-marine deposits have received diverse terminology (see Gaete et al., 2009 for a review). In the present study, we refer to the transition to fully continental materials deposited from the early Maastrichtian to the Thanetian as the Tremp Formation. This formation was divided into four lithological units by Rosell, Linares & Llompart (2001), which are from the base to the top: (1) a transitional ‘grey unit’ (marls, coals, limestones, and sandstones), (2) a fluvial ‘lower red unit’ (mudstones, sandstones, oncoids, and paleosols), (3) the lacustrine ‘Vallcebre limestone’ and laterally equivalent strata, and (4) a fluvial ‘upper red unit’ (mudstones, sandstones, conglomerates and limestones).

Although the Casa Fabà site is mostly covered by abundant vegetation, the outcrop consists of a surface of about 4 m2 of a sandstone layer with carbonate matrix inter-bedded between grey marl strata. These sediments are characteristic of the ‘lower red unit’. The occurrence of the charophyte Microchara punctata in those marl deposits would indicate a Maastrichtian age according to recent results of Vicente et al. (2015). These authors described a local Microchara punctata biozone ranging from the middle part of chron C31r to the lower part of chron C30n in the Vallcebre Basin. These results concur with the stratigraphic and biostratigraphic data of the site, which indicate an early Maastrichtian age, within the C31r chron (Riera et al., 2009; Díez-Canseco et al., 2014).

Material and Methods

Material

The recovered material was found in a 2 m2-area (Fig. 1C) including both cranial and postcranial elements (Figs. 2–6). Because no duplication of bones existed, and bones are connected or coherent in size, we consider the specimen to be a single individual. The skull is represented by the left premaxilla, a fragment of the right dentary, the right jugal and quadratojugal, most of the skull-table, and a damaged fragment of the jaw. The postcranial skeleton is also preserved and includes a right scapula, a fragmentary right humerus, a complete right ulna, a right dorsal rib, a proximal part of an indeterminate rib, an anterior dorsal vertebra and three lumbar vertebrae.

Figure 2 Skull of Allodaposuchus hulki sp. nov.

(MCD5139) and interpretative diagrams in (A) dorsal, (B) ventral, (C) caudal, and (D) left lateral view. Abbreviations: bc, basicranium; ctp, cavum tympanicum propium; cqp, cranioquadrate passage; ex, exoccipital; f, frontal; fa, foramen aëreum; fm, foramen magnum; fo, foramen; gef, groove for ear flap; la, lacrimal; lhc, lateral hemicondyle; ls, laterosphenoid; mhc, medial hemicondyle; olt, olfactory track; orb, orbit; p, parietal; pf, prefrontal; po, postorbital; ptf, postemporal fenestrae; q, quadrate; so, supraoccipital; sq, squamosal; stf, supratemporal fenestra. A and B are muscle scars on the quadrate.

Figure 3 Cranio-mandibular elements of Allodaposuchus hulki and interpretative diagrams.

Left premaxilla (MCD4763) in (A) dorsal, and (B) ventral view. Right dentary fragment (MCD5134) in (C) labial view. Right pair jugal-quadratojugal (MCD5129) in (D) dorsolateral and (E) ventromedial view. Indeterminate jaw fragment (MCD4758a) in (F) labial and (G) mesial view. Reconstruction of the skull of Allodaposuchus hulki (H). Abbreviation: al, alveoli; bpob, base of the postorbital bar; dal, dentary alveolus; dct, dentary caniniform tooth; drt, dentary replacement tooth; fo, foramen; ids, interdentary septum; if, incisive foramen; j, jugal; j-m, jugal-maxilla suture; mjf, medial jugal foramen; op, occlusion pits; pmal, premaxillary alveoli; pm-m, premaxilla-maxilla suture; pmt, premaxillary tooth; pm-sy, premaxilla symphysis; qj, quadratojugal; qj-q, quadratojugal-quadrate suture; qjs, quadratojugas spine; t, tooth.

Figure 4 Appendicular forelimb elements of Allodaposuchus hulki and interpretative diagrams.

Right scapula (MCD4765) in (A) medial, (B) anterior, (C) lateral, and (D) ventral view. Right humerus (MCD4758b) in (E) ventral, (F) medial, (G) dorsal, and (H) lateral view. Right ulna (MCD4760) in (I) lateral, (J) medial, (K) proximal, and (L) distal view. Abbreviation: ap, anterior process; dacu, distal anterior condyle of the ulna; dc, deltoid crest; dpc, deltopectoral crest; dpcu, distal posterior condyle of the ulna; fo, foramen; gl, glenoid; glf, glenoid fossa; lap, lateral anterior process; map, medial anterior process; mc, medial crest; mgr, medial groove; ol, olecranon; pcd, posterior circular depression; sb, scapular blade; sc-co, scapula-coracoid suture.

Figure 5 Muscular map of the forelimb bones of Allodaposuchus hulki.

Right scapula (MCD4765; A-medial, B-anterior, C-lateral), right humerus (MCD4758b, D-ventral, E-medial, F-dorsal, G-lateral), right ulna (MCD 4760, H-lateral, I-medial); and reconstruction of the anterior limb configuration (J). Muscle origins are indicated in pink and insertions in blue. Abbreviations: br, M. brachialis; cbd, M. coracobrahialis brevis dorsalis; dc, M. deltoideus clavicularis; DC, deltoid crest; ds, M. deltoideus scapularis; ecrd-pu, M. extensor carpi radialis brevis-pars ulnaris; fdl, M. flexor digitorum longus; fu, M. flexor ulnaris; hr, M. humeroradialis; ld, M. latissum dorsi; ls, M. levator scapulae; pq, M. pronator quadratus; shc, M. scapulohumeralis caudalis; sci, M. supracoracoideus intermedius; ss, M. subscapularis; svt, M. serratus ventralis thoracis; tb, M. triceps brachii; tbc, M. triceps brevis caudalis; tbi, M. triceps brevis intermedius; tlc, M. triceps longus caudalis; tm, M. teres major; tll, M. triceps longus lateralis.

Figure 6 Axial elements of Allodaposuchus hulki and interpretative diagrams.

Anterior dorsal vertebra (MCD5131) in (A) anterior, (B) posterior, (C) dorsal, and (D) left lateral view; first lumbar vertebra (MCD5136) in (E) anterior, (F) posterior, (G) dorsal, and (H) left lateral view; second lumbar vertebra (MCD4769) in (I) anterior, (J) posterior, (K) dorsal, and (L) left lateral view; third lumbar vertebra (MCD5126) in (M) anterior, (N) posterior, (O) dorsal, and (P) left lateral view. Abbreviations: aas, anterior articular surface; di, diapophysis; fo, foramen; hy, hypapophysis; nc, neural canal; ns, neural spine; par, parapophysis; poc, posterior condyle; poz, postzygapophysis; prz, prezygapophysis; psf, postspinal fossa.

The holotype of Allodaposuchus palustris (Blanco et al., 2014) and several extant crocodile skeletons were used as material of comparison, including one specimen of Crocodylus niloticus Laurenti, 1768 (MZB2003-1423), two of Alligator mississippiensis Daudin, 1802 (MZB2006-0613, MZB92-0231) and one Osteolaemus tetraspis Cope, 1861 (MZB2006-0039). In addition, we gathered both cranial and postcranial information from the literature about extant and extinct crocodylomorphs: Crocodylus acutus Cuvier, 1807 (see Mook, 1921), Sebecus icaeorhinus Simpson, 1937 (see Pol et al., 2012), Allodaposuchus precedens (Buscalioni et al., 2001; Delfino et al., 2008) and Allodaposuchus subjuniperus (Puértolas-Pascual, Canudo & Moreno-Azanza, 2014).

Anatomical nomenclature

The description of the cranial osteology of the new material follows the terminology used in those works concerning the genus Allodaposuchus (Buscalioni et al., 2001; Delfino et al., 2008; Blanco et al., 2014; Puértolas-Pascual, Canudo & Moreno-Azanza, 2014), whereas postcranial elements are described following Mook (1921) and Brochu (2011). In addition, the terminology used to describe appendicular myological features is according to Meers (2003).

Phylogenetic analyses

Phylogenetic relationships of the specimen from Casa Fabà were explored using the dataset of Brochu (2011). However modifications in some operational taxonomic units (OTUs) and characters were carried out (see Supplemental Information S1).

The entire dataset resulted in 86 OTUs coded for a total of 181 craniodental and postcranial characters (see Supplemental Information S2). The taxon Bernissartia fagesii Dollo, 1883 was used as outgroup. The dataset was analysed with TNT v1.1 (Willi Hennig Society Edition, Goloboff, Farris & Nixon, 2008). Tree-space was explored using a heuristic search algorithm (traditional search method) with tree-bisection-reconnection branch swapping and 1,000 random addition replicates holding 10 most parsimonious trees for each replicate. All characters were equally weighted and multistate characters were unordered. Bremer supports and bootstrap frequencies (1,000 bootstrap replicates searched) were used to assess the robustness of the nodes.

Inner structural exploration

A Computed Tomography scanner (CT-scan) was used to explore the morphology and the inner structure of the cranial elements. The remains were scanned by multi-detector computer tomography (Sensation 16; Siemens) at Hospital Universitari Mútua de Terrassa (Terrassa, Spain). The material was scanned at 140 kV and 300 mA with an output of 512 × 512 pixels per slice, with an interslice space of 0.3 mm obtaining a pixel size of 0.586 mm and processed with the Avizo 7.0 software (FEI VSG Company). This process allows the recognition of inner cranial structures.

Nomenclatural acts

The electronic version of this article in Portable Document Format (PDF) will represent a published work according to the International Commission on Zoological Nomenclature (ICZN), and hence the new names contained in the electronic version are effectively published under that Code from the electronic edition alone. This published work and the nomenclatural acts it contains have been registered in ZooBank, the online registration system for the ICZN. The ZooBank LSIDs (Life Science Identifiers) can be resolved and the associated information viewed through any standard web browser by appending the LSID to the prefix “http://zoobank.org/”. The LSID for this publication is: urn:lsid:zoobank.org:pub:3735BA19-C38F-4F6E-93A5-3D302580E818. The online version of this work is archived and available from the following digital repositories: PeerJ, PubMed Central and CLOCKSS.

Systematic Paleontology

Order CROCODYLIFORMES Hay, 1930 (sensu Benton & Clark, 1988),	
Suborder EUSUCHIA Huxley, 1875,	
Unranked CROCODYLIA Gmelin, 1789 (sensu Benton & Clark, 1988),	
Genus Allodaposuchus Nopcsa, 1928,	
Allodaposuchus hulki sp. nov.	
	
urn:lsid:zoobank.org:act: 267AADFA-AD84-45F4-B195-D08E174559CC	

(Figs. 2–6).

Etymology:hulki, from the character of Marvel, Hulk; due to the strong muscle attachments of the bones.

Differential diagnosis:Allodaposuchus differs from all other Eusuchians by the presence of the canalis quadratosquamosoexoccipitalis, or cranioquadrate passage, laterally open and represented by a sulcus (broader than in Hylaeochampsa vectiana Owen, 1874). Allodaposuchus hulki differs with Allodaposuchus palustris in having a linear frontoparietal suture, a prominent boss on paraoccipital process of the exoccipital, a small foramen aëreum, and lacking false-ziphodont teeth. Allodaposuchus hulki differs from A. subjuniperus by having the incisive foramen abutting the premaxillary tooth row, located between the first and second alveoli, the premaxillary-maxillary suture does not reach the incisive foramen, external naris opened in aterodorsal direction, no elevation around the rim of the external naris, absence of interorbital ridge, a very large medial jugal foramen, quadratojugal spine nearly absent and located near of the ventral angle in the infratemporal fenestra, medial articular hemicondyle of the quadrate without ventral expansion, and teeth without longitudinal grooves in the lingual side. Allodaposuchus hulki differs from A. precedens by having the premaxilla wider than long with four teeth positions, the third being the largest, a smaller and keyhole-shaped external naris, no elevation around the rim of the external naris, incisive foramen located between the first and second alveoli, dermal bones of skull roof overhanging supratemporal fenestra rim, two crests in the ventral surface of the quadrate without association of any tubercle, and capitate processes of laterosphenoid anteroposteriorly oriented, and teeth with smooth enamel.

Allodaposuchus hulki shows the following autapomorphies: Quadratojugal does not extend along the infratemporal fenestra; absence of fossa or shelf at anteromedial corner of the supratemporal fenestra; teeth bear smooth enamel, low-developed mesial and posterior carinae, and absence of longitudinal grooves in the lingual side.

Aside the previous characters, A. hulki has the following ambiguous autapomorphies: Spine of quadratojugal significantly reduced; no ridge surrounds the foramen aëreum; anterolateral, anteromedial and olecranon processes of the ulna well developed; ulnar shaft lateromedially compressed with lateral and medial grooves; distal condyles of the ulna turned lateroposteriorly, causing a lateral crest in the shaft. These features are ambiguous autapomorphies because these characters cannot be scored in other species of Allodaposuchus. Additionally, variability might affect the ridge surrounding the ridge surrounding the foramen aëreum. New discoveries may reveal whether they are autapomorphies of A. hulki or synapomorphies of the genus.

Holotype: MCD4757 (rib fragment), MCD4758a (jaw fragment), MCD4758b (right humerus), MCD4760 (right ulna), MCD4763 (left premaxilla), MCD4765 (right scapula), MCD5127 (dorsal rib), MCD5129 (right jugal and quadratojugal), MCD5131 (dorsal vertebra), MCD5134 (dentary fragment), MCD5139 (skull-table, exoccipitalis and left quadrate), MCD4769, MCD5126 and MCD5136 (lumbar vertebrae).

Locality, age and horizon: Casa Fabà site, Tremp Basin (NE Spain); lower part of the ‘lower red unit’ of the Tremp Fm; C31r of the early Maastrichtian (Late Cretaceous).

Description

Cranial skeleton

The cranial remains consist of an isolated left premaxilla (MCD4763), jaw fragments (MCD5134 and MCD4758a), an isolated right jugal and quadratojugal (MCD5129), and a posterior cranial fragment (MCD5139) that preserves frontal, left prefrontal, parietal, both squamosals, postorbitals, exooccipitals, left quadrate and laterosphenoid in connection (Figs. 2 and 3). The preserved portion of the skull table is markedly medially concave, with nearly horizontal sides and displays roughly straight margins. The supratemporal fenestrae are filled with sediment (Figs. 2A–2B). We estimated a total width ranging from 27 cm (from both lateral hemicondyles of the quadrates) to 34 cm (from both lateral edges of the quadratojugals).

Cranial openings

The external naris is undivided and keyhole-shaped (Figs. 3A–3B). It is 3.2 cm wide and opens in the antero-dorsal direction. In ventral view, there is a small and subcircular incisive foramen (1.7 cm wide), the anterior rim of which is located between the first and second alveoli (Fig. 3B). The medial margin of the left orbit is preserved, being able to interpret its general morphology. The orbits are relatively wide and short, rounded with their rostromedial margin being somewhat elevated. Supratemporal fenestrae are subcircular in shape (4.5 cm maximum width) and filled with sediment. There is no fossa or shelf at the anteromedial corner of the supratemporal fenestrae. The otic aperture is developed between the squamosal, quadrate and exoccipital, and the cranioquadrate passage forms a caudolaterally open sulcus called canalis quadratosquamosoexoccipitalis (Buscalioni et al., 2001; Delfino et al., 2008). The squamosal and the quadrate are not in contact posteriorly to the otic opening.

Premaxilla

The premaxilla is nearly complete and characterized by its robust appearance. The premaxilla is rounded, and wider than long. Its posterodorsal margin is slightly eroded; thus, the presence of a notch or pit in the palatal side of premaxillary-maxillary suture, or the length of the premaxillary process, could not be confirmed. It contacts the maxilla posterolaterally, and probably the nasals medially (Figs. 3A–3B). There is no elevation along the lateral rim of the naris, and none seems to occur posteriorly. The naris opens flush with the dorsal surface of the premaxilla, without the development of any lateral notch. The internal cavity of the naris shows a large foramen in the rostral-most portion of the left surface, and several longitudinal ridges caudally to the incisive foramen, probably for soft tissues or muscle attachment. On the palatal surface, the premaxillary-maxillary suture does not abut the posterior margin of the incisive foramen, with the latter being completely included within the premaxilla. There are four premaxillary alveoli, but only the fourth tooth root is preserved within its socked. The first premaxillary alveolus is the smallest in size, while the second and fourth are similar in size, and the third is the largest (Fig. 3B). There is one occlusal pit between the first and second alveoli, and another between the second and third. No pit is present between the third and fourth alveoli.

Jugal

A complete right jugal bone, not preserving the postorbital bar, has been recovered (Figs. 3D–3E). It displays an elongated morphology and shows an ornamented external surface. Rostral and caudal edges of the jugal are respectively lateromedial turned, keeping an approximately constant lateromedial width. Along the orbit, the jugal dorsal margin is slightly elevated for contact with the lacrimal, making a rounded ventral margin of the orbit. The postorbital bar is lost, but insertion on the jugal can be defined as ‘inset’. In lingual view, an uncommonly large medial jugal foramen can be observed rostrally to the postorbital bar insertion. Another smaller foramen is also present rostrally to the former. Ventrally to these foramina, the articular surface with the maxilla begins and continues to the rostral edge.

Quadratojugal

Quadratojugal is a short and wide bone, forming the posterior angle of the infratemporal fenestra (Figs. 3D–3E). It does not bear any process along the lower temporal bar. It also does not extend to the superior angle of the infratemporal fenestra. The quadratojugal spine is nearly absent and low in position, near to the posterior angle of infratemporal fenestra. In the lateral side, jugal and quadratojugal bear the same ornamentation pattern as the skull table.

Quadrate

Only the left quadrate is complete, being part of MCD5139 (Fig. 2). In lateral view, the quadrate contacts the squamosal rostrally, and the exoccipital caudally, forming the anteroventral margin of the external otic aperture and the ventral limit of the cranioquadrate passage. The quadrate also contacts the postorbital ventrally to the skull table, in the dorsal margin of the infratemporal fenestra. In dorsal view, the quadrate is short caudally to the paroccipital process of the exoccipital bone. Both articular hemicondyles are similar in size, although the medial hemicondyle is slightly smaller and ventrally deflected (Fig. 2C). From the posteroventral corner of the otic aperture, a soft sulcus passes along the quadratoexoccipital suture, in the posterolateral direction, parallel to the cranioquadrate passage, and ends abruptly. The foramen aëreum is small and located on the dorsal surface, close to the medial edge of the quadrate. No ridge surrounds the foramen aëreum. In ventral view, there are two well-marked crests corresponding to the muscle scars A and B of Iordansky (1973), without the association of any tubercle (Fig. 2B). In contrast, the right quadrate is just broken ventrally to the otic aperture, showing the otic canal, also known as the cavum tympanicum propium, filled with sediment.

Frontal

The frontal forms the posteromedial corners of the orbits and the anteromedial corners of the supratemporal fenestrae. It contacts the postorbital laterodorsally and the parietal caudally (Fig. 2A). The frontal prevents contact between postorbital and parietal. The frontoparietal suture is nearly linear and enters the rostromedial margins of the supratemporal fenestrae. The dorsal surface of the frontal is markedly ornamented by subcircular pits that can reach 3 mm in diameter. The main body of the frontal is strongly concave at the centre of the dorsal surface, and the orbital margins are upturned. No interorbital ridge is present. The anterior process of the frontal is not preserved. At least part of the left prefrontal is also preserved in MCD5139 (Figs. 2A–2B).

Parietal

It contacts the frontal anteriorly and the squamosal laterally (Fig. 2A). There is no contact between the parietal and the postorbital in dorsal view. Contact with the supraoccipital could not be assessed due to preservation reasons. The parietal is longer than wide and displays a marked ornamentation on the dorsal surface, consisting of the same subcircular depressions present in the rest of the skull bones. The parietal is medially concave, as part of the general concavity of the skull table. A recess in the parietal communicates with the pneumatic system.

Postorbital

It contacts the squamosal posteriorly and the frontal anteromedially. In MCD5139, both postorbitals are longer than wide, displaying a well-curved contour in dorsal view (Fig. 2A). Postorbital constitutes the anterolateral corner of the supratemporal fenestra, conferring rounded anterior edges to the skull table. It also forms the posterior orbital margin and the anterodorsal corner of the infratemporal fenestra. The ornamentation is the same as that of the rest of the skull table.

Squamosal

The squamosal is a triangular-shaped bone which contacts the postorbital anteriorly, the parietal medially, the quadrate anteroventrally, and the exoccipital posteroventrally, constituting the cranioquadrate passage (Fig. 2D). Posteriorly to the passage, the quadrate and the squamosal are separated by the exoccipital. In lateral view, two rims delimited a longitudinal groove for external ear valve musculature. These dorsal and ventral rims are parallel. In dorsal view, the suture between the squamosal and the postorbital is very posteriorly situated, reaching the caudal-most part of the lateral margin of the supratemporal fenestra. The squamosal develops a significantly posterolateral extension resulting in a horizontal margin of the skull table. In occipital view, the squamosal slopes ventrolaterally over the exoccipital, but the squamosal does not laterally surpass the paroccipital process of the exoccipital.

Supraoccipital

The skull table is damaged coinciding with the supraoccipital location; thus, the morphology of the supraoccipital and its relationships with other bones could not be assessed with confidence.

Exoccipital

It occupies most of the occipital surface, contacting the squamosal dorsally, and the quadrate lateroventrally (Fig. 2). The exoccipital conforms the caudoventral margin of the cranioquadrate passage. The paroccipital process does not extend much laterally, ending in the medial quadrate branch. In occipital view, the exoccipital shows a very prominent boss on the paraoccipital process. The foramen magnum is relatively preserved, but the ventral edges of exoccipitals are broken, and the suture with the basioccipital is missing.

Laterosphenoid

In palatal view, the laterosphenoid is situated medially in the braincase, between the supratemporal fenestrae (Fig. 2B). It contacts frontal rostrally and postorbitals laterally, conforming the rostromedial margin of supratemporal fenestra in ventral view. The capitate process of the laterosphenoid is anteroposteriorly oriented.

Dentary

Only a right fragment of the anterior part of the dentary was recovered (MCD5134), which is very fragmentary and incomplete (Fig. 3C). According to the medial curvature of the bone, we have interpreted this as the rostral-most portion of the right dentary, bearing from the first to the fourth alveoli. The lingual surface of the bone is not preserved, showing the alveoli in a section. Only two teeth are preserved in situ, the second and fourth, projecting anterodorsally. In the second alveolus, the functional tooth is lost, but there is a replacement non-erupted tooth. In contrast, the fourth tooth is a caniniform.

MCD4758a is an indeterminate fragment of the jaw. For preservation reasons, it is not possible to elucidate its position along the jaw. This fragment preserves three alveoli, bearing one erupted tooth (Figs. 3F–3G).

Dentition

The whole specimen preserves four teeth (one root in the premaxilla, two teeth in the dentary, and one tooth in MCD4758a). The teeth are circular in section and the tooth crowns are slightly blunt. Enamel lacks ornamentation, but several longitudinal ridges appear in the most basal portion of the crown in the lingual side. Anterior and posterior keels are poorly developed, and there are no longitudinal grooves next to the keels in the lingual side. These ridges are weak in MCD4758a, and more developed in the caniniform tooth of MCD5134. Replacement tooth of MCD5134 lacks ridges, probably due to being a non-erupted tooth (Fig. 3C).

Postcranial skeleton

Recovered postcranial skeleton of Allodaposuchus hulki is composed of a distal end of the right scapula (MCD4765), right humerus (MCD4758b), right ulna (MCD4760), two dorsal ribs (MCD4757 and MCD5127) and five vertebrae (one dorsal, three lumbar, and fragments of an indeterminate one, MCD5131, MCD5136, MCD4769, MCD5126 and MCD5125, respectively).

Scapula

Only the ventral edge of the scapula is preserved, showing the glenoid fossa, the deltoid crest, the anterior process of the scapula and part of the scapular blade (Figs. 4A–4D). The scapular blade is constricted at its beginning, and seems to flare dorsally. In posterior view, the scapular blade is sinuous. The glenoid is oriented posterolaterally. In lateral view, the posterior end of the scapular blade is straight. The lateral surface of the scapula bears several rugose areas (Fig. 4A). A rugose zone for the insertion of the M. serratus ventralis thoracis (Meers, 2003) is situated in the posterior edge of the scapular blade. In the same side, but in a more ventral position where the blade is constricted, another rugose area evidences the origins of the M. scapulohumeralis caudalis, and just dorsal to supraglenoid buttress a highly-developed rugosity constitutes the origin of the M. triceps longus lateralis (Meers, 2003). The anterior process of the scapula bears a wide deltoid crest. This crest is the origin of the M. coracobrachialis brevis dorsalis and the M. deltoideus clavicularis. Ventrally to the deltoid crest, a softer rugose area evidences the origin of the M. supracoracoideus intermedius (Figs. 4A–5C; Meers, 2003). In medial view, the scapula also shows several rugose areas for muscle attachment (Fig. 4A). The medial surface of the blade shows the origin of M. subscapularis, and M. triceps longus caudalis (Fig. 5A; Meers, 2003). In this view, a small foramen is present at the base of the scapular blade. The anterior process of the scapula also bears a soft rugose area for the origin of M. supracoracoideus longus (Meers, 2003). In the ventral side, the scapula-coracoid facet is present, and is broader caudally (Fig. 4D). Coracoid is not fused to the scapula. The sutural surface is teardrop-shaped, and the lateromedial width in the posterior portion is much greater than the anterior one.

Humerus

A right humeral diaphysis was recovered (Figs. 4E–4H). The proximal articular surface is eroded. The anterior tuberosity and the humeral head seem to have the same height, and the posterior tuberosity seems to be slightly distally positioned. The deltopectoral crest is missing because of the preservation. In lateral view, the proximal portion of the shaft is dorsally concave, and its distal portion is ventrally turned. The lateral surface of the humerus is slightly concave. In turn, the medial surface of the humerus is slightly convex. The shaft is broken proximally to the distal condyles.

Like the scapula, the humerus also shows several rugous areas around the bone. In posterior view (Fig. 4G), a wide rugous area is situated caudally to the condyles, corresponding to the insertion of the M. scapulohumeralis caudalis (Figs. 5D–5G; Meers, 2003). The dorsal surface bears a single insertion scar for M. teres major and M. dorsalis scapulae (according to Brochu, 2011), or M. teres major and M. latissimus dorsi (according to Meers, 2003). The shaft shows small and soft ridges completely occupying the dorsal surface. These ridges correspond at least to the origin of the M. triceps brevis intermedius, and possibly the M. triceps brevis cranialis (Meers, 2003), but limits between both muscles are not distinguished. The origins of M. triceps brevis intermedius extends to medial side of the shaft (Figs. 5D–5G). The lateral surface of the shaft shows part of the origins of M. triceps brevis intermedius, and the origin of the M. humeroradialis (Meers, 2003). The deltopectoral crest is lost, lacking most of the M. deltoideus clavicularis insertion (Meers, 2003). A small crest (lineae intermuscularis humeroradialis-brachialis) is situated laterally to the insertion of M. teres major. In ventral view (Figs. 5D–5G), descriptions for muscle attachment in the proximal epiphysis could not be made for preservation reasons. However, the shaft shows the end of the origin of M. triceps brevis intermedius in medial margin, and the end of the origin of M. humeroradialis in the lateral one. At the middle of the shaft, the origin of M. brachialis is present (Meers, 2003), distally to the deltopectoral crest location.

Ulna

The right ulna is one of the best-preserved bones (Figs. 4H–4K). The proximal end of the ulna is anterioposteriorly expanded. In proximal view, it is triangular-shaped, with a very prominent vertex. The olecranon process is acute, and the articular surface for the radius is concave. There is a very sharp process medially to the articular surface for the radius. The shaft is compressed mediolaterally, and shallow grooves could be discerned in the medial and lateral sides. The distal half of the shaft is prominently oriented anteriorly. Both distal condyles are compressed and anterolateral to mediocaudally oriented. As a result of this torsion, there is a ridge in the lateral side of the distal end of the ulna.

The ulna also shows several muscle attachments. A rugosity abuts the olecranon process in caudal, lateral and medial views, and evidences the insertion of M. triceps brachi (Meers, 2003). In lateral view (Fig. 4H), the M. flexor ulnaris originates in a groove situated just distally to the sharp process of the articular surface of the radius. M. flexor ulnaris disposes over the lateral ridge of the ulna. Rostrally to it, a very soft ridge conforms the origin of M. extensor carpi radialis brevis—pars ulnaris, close to the anterior margin and facing to the radius (Fig. 5H). In the medial side (Fig. 4I), other groove supports the origin of M. pronator quadratus (Meers, 2003). A small crest is situated caudally to this groove, in addition to other soft ridges, which comprise the origin of M. flexor digitorium longus (Fig. 5I; Meers, 2003). There are also very rough areas laterally and medially to the distal condyles for ligament attachments, with the lateral one being more developed than the medial.

Vertebrae

The preserved vertebrae have been sorted based on the presence of the keels and the position of the parapophyses, according to Mook (1921). The fourth dorsal vertebrae (MCD5131) and the first (MCD5136), second (MCD4769) and third (MCD5126) lumbar have been recovered. All vertebrae are strongly procoelus.

MCD5131 (Figs. 6A–6D) is broken at the right side, and prezygapophyses, right diapophysis and parapophysis, and neural spine are lost. Postzygapophyses are elongated lateromedially and short anteroposteriorly, its articular surface faces ventrally and is lateromedially inclined. The left parapophysis is located at the base of diapophysis. The diapophysis is elongated and dorsolaterally oriented, but thin anteroposteriorly. The centrum and the neural arch are completely fused. The centrum is relatively short anteroposteriorly. A well-developed hypapophyseal keel is present ventrally to the centrum. Wide striated areas can be observed at the base of the neural arch, under the diapophyses and postzygapophyses, in the ventral side of the centrum, and dorsally to the neural arch, between the diapophysis, post- and prezygapophysis, and neural spine. These areas are consistent with a strong muscle attachment.

Lumbar vertebrae (MCD5136, MCD4769 and MCD5126; Figs. 6E–6P) are partially broken or eroded. Pre- and postzygapophyses are broader than those of the dorsal, and progressively wider from the first to the third lumbar. Neural spines are partially broken, but they seem to be wide anteroposteriorly and low. Transverse processes are horizontally oriented, and progressively decrease in height with respect to the centrum. They are laterally shorter and anteroposteriorly wider from the first to the third lumbar. There is a longitudinal groove ventrally to the centrum in all lumbar vertebrae. Like the dorsal vertebrae, wide striated areas are present laterally to the neural arch, across the diapophyses, in the ventral side of the centrum, and between the diapophysis, post- and prezygapophysis, and neural spine.

Ribs

Two partial ribs were recovered. One of them comprises only the shaft of a right dorsal rib (MCD5127) lacking its capitular and tubercular processes. It is elongated (19.2 cm long) and lateromedially compressed. In section, it is teardrop-shaped, with its thinner margin cranially, forming an anterior crest of the shaft. Soft ridges for muscle or ligaments attachments are present lateral and medially in the distal-most portion of the shaft. In turn, MCD4757 consists of the capitular and tubercular processes of an anterior-most left dorsal rib, which lacks the shaft.

Comparative anatomy

Cranial remains are comparable to other species of Allodaposuchus, especially to the nearly complete skulls of A. precedens and A. subjuniperus (Delfino et al., 2008; Puértolas-Pascual, Canudo & Moreno-Azanza, 2014; respectively). However, postcranial remains are only comparable to A. palustris (Blanco et al., 2014) due to the absence of published material of the others.

Several characters of the premaxilla distinguish Allodaposuchus hulki from the other species of the genus. The external naris opens in the antero-dorsal direction like A. precedens. In turn, naris of A. subjuniperus opens in dorsal direction. The external naris is significantly smaller than A. precedens and keyhole-shaped in A. hulki. Unlike A. precedens and A. subjuniperus, there is no elevation along the rim of the external naris. However, no lateral notch develops on the dorsal surface of premaxilla next to the naris opening, like A. precedens and A. subjuniperus. In palatal view, the incisive foramen is located more anteriorly than A. precedens and A. subjuniperus. In A. hulki the anterior rim of the incisive foramen is located between the first and second alveoli, whereas in A. subjuniperus, it reaches the third premaxillary alveolus. The premaxilla is wider than long, like A. subjuniperus. However, unlike A. subjuniperus, the premaxillary-maxillary suture does not reach the posterior margin of the incisive foramen, like A. precedens. The number of premaxillary alveoli is four, with the third alveolus being the largest, like A. subjuniperus. The premaxilla of A. precedens shows five teeth, and the fourth is the largest. Moreover, A. hulki shows a pattern of occlusal pits, different to A. precedens and A. subjuniperus: there is one occlusal pit between the first and second alveoli, another between the second and third, and no pit between the third and fourth alveoli. In turn, A. precedens shows one occlusal pit between the third and the fourth, and another between the fourth and the fifth alveoli, whereas A. subjuniperus shows only a large diastema between the first and second alveoli.

In addition, the skull table of A. hulki also shows differences from the other species of the genus. The main body of the frontal is concave medially, like A. palustris and A. precedens, and the orbital margins are upturned. However, this concavity is strongly marked in A. hulki, but only slightly in A. palustris and A. precedens. In contrast, A. subjuniperus shows a practically flat frontal with a low transverse interorbital ridge at the beginning of the anterior process. Allodaposuchus palustris also shows the interorbital ridge. Nevertheless, no interorbital ridge is present in A. hulki and A. precedens. The orbits of A. hulki are wide and short, like A. precedens and A. subjuniperus. In contrast, A. palustris shows relatively large and elongated orbits. However, the four species of Allodaposuchus have an elevated rostromedial margin of the orbits. The frontoparietal suture of A. hulki is nearly linear, like A. subjuniperus and A. precedens. Allodaposuchus palustris is the only ‘allodaposuchian’ that shows a concavo-convex frontoparietal suture. Additionally, A. hulki and A. palustris are the two species that do not show a fossa or shelf around the supratemporal fenestrae.

Like A. precedens, both articular hemicondyles of the quadrate of A. hulki are similar in size, although the medial hemicondyle is slightly smaller and ventrally deflected. In turn, A. subjuniperus shows a ventral expansion in the medial hemicondyle. The foramen aëreum of A. hulki is small, like A. precedens and A. subjuniperus, but large in A. palustris. This foramen is located on the dorsal surface, close to the medial edge of the quadrate, in all species of Allodaposuchus. However, in A. hulki, no ridge surrounds the foramen aëreum, unlike the other three species of Allodaposuchus. Like A. subjuniperus, in ventral view, there are two muscle scars of Iordansky (1973), without the association of any tubercle. Allodaposuchus precedens only shows one crest ending abruptly and forming a small tubercle, and A. palustris does not show any crest.

The exoccipital of A. hulki, A. subjuniperus, and A. precedens bears a very prominent boss on the paraoccipital process. Allodaposuchus palustris is the only ‘allodaposuchian’ that does not show this boss.

In ventral view, the capitate process of the laterosphenoid is anteroposteriorly oriented in A. hulki and A. subjuniperus, but is laterally oriented in A. precedens.

The quadratojugal of A. hulki shows a characteristic shape that represents a few autapomorphies of this taxon. Unlike A. precedens and A. subjuniperus, the quadratojugal does not extend to the superior angle of the infratemporal fenestra. Moreover, the quadratojugal spine is almost absent and near the ventral angle in the fenestra infratemporal. Allodaposuchus precedens and A. subjuniperus show the quadratojugal spine in a higher position in the fenestra. This spine is highly developed in A. subjuniperus.

The dentition of A. hulki also shows several characters that distinguish this form from the other species of the genus. In A. hulki, the enamel lacks ornamentation, both anterior and posterior carinae are poorly developed, and there are no longitudinal grooves in the lingual side. In contrast, A. palustris and A. precedens show ornamented enamel with well-developed carinae, whereas A. subjuniperus and A. palustris bear well-marked longitudinal grooves in the lingual side.

Concerning the axial skeleton, all recovered vertebrae are procoelous. This character clearly indicates a Eusuchian condition (Salisbury et al., 2006). Preserved dorsal and lumbar vertebrae are similar in shape to those of Crocodylus acutus, Crocodylus niloticus, Alligator mississipiensis, Osteolaemus tetraspis and Allodaposuchus palustris. However, all of the vertebrae of A. hulki show wide areas for muscle or ligament attachment, which are absent in those of A. palustris.

The appendicular skeleton is poorly comparable to A. palustris. This specimen does not preserve the scapula and ulna, but a fragmentary humerus, radius and hindlimb bones (Blanco et al., 2014). The scapula of A. hulki is similar in shape to those of extant crocodylians. The scapular blade shows a constriction dorsally to the glenoid and anterior process, and flares dorsally. However, in caudal view, the glenoid is more conspicuous and the scapular blade is more sinuous than in extant taxa. In addition, the scapula of A. hulki bears more developed scars for the origin of M. scapulohumeralis caudalis, M. triceps longus lateralis, M. supracoracoideus intermedius and M. supracoracoideus longus than in the other taxa compared. The humerus of A. hulki is clearly more robust and also shows more developed origins of M. humeroradialis and M. brachialis than in A. palustris and other extant taxa. In the latest taxa, the humeral surface is mainly smooth, but in A. hulki it is completely covered by soft ridges for muscle attachments. However, the humerus of A. hulki resembles those of the compared taxa in shape. In contrast, the ulna of A. hulki largely differs from the morphology of the extant crocodylians. The proximal epiphysis of the ulna of A. hulki has three well-developed processes (the olecranon directed posteriorly, an anteromedial process, and an anterolateral process). These processes are weakly developed in extant crocodylians, but well-developed in Sebecus and other terrestrial notosuchians (Pol et al., 2012; and references therein). Nevertheless, the general morphology of the proximal surface of the ulna is different between those fossil taxa and the new taxon of Allodaposuchus. The shaft of this bone is lateromedially compressed in A. hulki, unlike the other compared taxa that show a subcircular shaft in cross-section. Additionally, the torsion of the distal condyles, the crest for the origin of M. flexor digitorium longus, and the lateral and medial grooves for the origin of M. flexor ulnaris and M. pronator quadratus, respectively, have not been seen in other taxa used for comparison.

Thus, given all of the factors discussed above, Allodaposuchus hulki seems to be morphologically intermediate to A. subjuniperus and A. precedens, with several unique characters. In addition, as noted by Blanco et al. (2014), the appendicular skeleton seems to be conservative in fossil ‘allodaposuchians’ and living taxa. Nevertheless, despite appendicular bones could not be directly compared between A. hulki and A. palustris, both ‘allodaposuchians’ show stronger muscle and ligament scars than in other taxa used for comparison.

Results

Endocranial configuration

As typical for archosaurs, the brain did not occupy the entire endocranial cavity (Witmer et al., 2008). The 3D reconstruction of the cranial fragment reveals the morphology of some parts of the olfactory bulbs, the cerebral hemispheres, cranial nerves, inner middle ear and tympanic recesses (Fig. 7).

Figure 7 Cranial endocast and pneumatic sinuses within the semi-transparent body skull of Allodaposuchus hulki.

MCD5139 derived from surface rendering of CT scan data in (A) dorsal, (B) ventral, (C) caudal, and (D) left lateral view. (E) Detail of the braincase and cranial nerves; the inner ear is removed because it obscured some endocast details. Abbreviations: adtr, anterodorsal tympanic recess; cer, cerebral hemisphere; cn, cranial nerves; cqp, craniocuadrate passage; ctp, cavum tympanicum propium; ctr, caudal tympanic recess; dls, dorsal line dural venous; dtr, dorsal tympanic recess; ie, inner ear; itr, intertympanic recess; ob, olfactory bulb; ot, olfactory track; qs, quadrate sinus; sps, sphenoparietal dural venous. Cranial nerve identification: III, oculomotor nerve canal; IV, trochlear nerve canal; V1, opthalmig nerve canal; V2, maxillary nerve canal; V3, mandibular nerve canal; VI, abducens nerve canal; XII, hypoglossal nerve canal.

The preserved portion of the endocast shows the typical sigmoid morphology. However, just the most anterior part of the olfactory bulbs are preserved, whereas the cerebral hemisphere is dorsally well preserved but lacking some ventral areas for preservation reasons. Some of the left side cranial nerves are recognisable, in particular the nerve V system, particularly V1, V2 and V3 as well as nerves III, IV and XII. The inner middle ear is present but badly preserved. Moreover, the low pixel resolution of the CT scan avoids assessing its morphology with confidence and is therefore not included in the figures (Fig. 7E).

Of particular interest are the tympanic recesses in the cranial fragment. On the one hand, the intertympanic recess is very developed. In extant crocodilians, such as Crocodylus johnstoni Krefft, 1873 (Witmer & Ridgely, 2008), the anterior part of the intertympanic recess presents a semicircular morphology, whereas in the described specimen, a complete circle morphology is found due to the development of an anterior portion. In connection with the intertympanic recess, there is a cavum tympanicum propium with a similar morphology to that in other extant crocodilians. However, on the other hand, the quadrate sinus is very wide in comparison with other extant crocodilians (Witmer et al., 2008; Witmer & Ridgely, 2008) in the reported specimen.

Phylogenetic relationships

The cladistic analysis resulted in 1,240 equally parsimonious cladograms of 600 steps (CI = 0.382; RI = 0.811; RC = 0.310), and includes the specimen from Casa Fabà in the genus Allodaposuchus. The strict consensus tree (Fig. 8) shows similar topology to the last hypothesis about Allodaposuchus emplacement (Blanco et al., 2014), in contrast to some previous works that suggest that Allodaposuchus is a sister taxa of hylaeochampsids (Buscalioni et al., 2001; Delfino et al., 2008; Puértolas-Pascual, Canudo & Moreno-Azanza, 2014) or derived alligatoroid (Martin, 2010). In the present analysis, as in Blanco et al. (2014), the clade ‘Allodaposuchia’ was included within Crocodylia, placed in a more derived position than Gavialoidea, and forming a polytomy with Borealosuchus, Planocraniidae and the clade Brevirostres (Crocodyloidea + Alligatoroidea). However, in the present analysis, the relationships between Arenysuchus and Allodaposuchus species were better resolved. In Blanco et al. (2014), A. palustris is the most basal ‘allodaposuchian’, whereas A. precedens, A. subjuniperus, and Arenysuchus form a polytomy. Our results suggest that A. precedens and the Eusuchian from Casa Fabà are more derived than A. subjuniperus and Arenysuchus. Both results agreed that A. palustris is the most basal crocodylian of the genus, and that Arenysuchus gascabadiolorum could be included within the clade ‘Allodaposuchia’. However, the relationships of Arenysuchus with this genus exceed the aim of this paper, and should be confirmed in detail in future works.

Figure 8 Resulting strict consensus cladogram illustrating the phylogenetic relationship of Allodaposuchus hulki and the basal position of ‘Allodaposuchia’ within Crocodylia.

Values above nodes represent bootstrap percentage, whereas values under nodes represent Bremer support.

Discussion

Phylogeny

The most parsimonious hypothesis obtained in our analyses suggests that the clade ‘Allodaposuchia’ belongs to Crocodylia (Fig. 8). Even though the Bremer and bootstrap values were low, the clade ‘Allodaposuchia’ has similar support to the other clades of Crocodylia. According to Blanco et al. (2014), the genus Allodaposuchus might represent more derived Eusuchian crocodylomorphs than previously thought (Buscalioni et al., 2001; Delfino et al., 2008; Narváez & Ortega, 2011; Puértolas-Pascual, Canudo & Moreno-Azanza, 2014) but not as derived as in Martin (2010). This hypothesis is supported by several cranial and vertebral characters. According to Brochu (1997), the inclusion of the clade ‘Allodaposuchia’ within Crocodylia would be supported by the following synapomorphies: (1) anterior dentary teeth project anterodorsally, (2) retroarticular process projects posterodorsally, (3) frontoparietal suture concavo-convex, (4) mature skull table with nearly horizontal sides, and long posterolateral squamosal rami along paraoccipital process, (5) exoccipital lacks boss on paraoccipital process, and (6) hypapophyseal keels are present on the eleventh vertebrae behind the atlas. Absence of the boss in the paraoccipital process, only in A. palustris, would be an ancestral state reverted in other members of the clade ‘Allodaposuchia’ (Blanco et al., 2014). The following synapomorphies related Allodaposuchus with Borealosuchus + Planocraniidae + Brevirostres (sensuBrochu, 1997): (1) slender postorbital bar, (2) ventral margin of postorbital bar inset from lateral jugal surface, (3) skull table surface planar at maturity, (4) frontoparietal suture concavo-convex, (5) neural arch of the axis lacking lateral processes (diapophyses), (6) wide posterior half of the axis neural spine, (7) axial hypapophysys without deep fork, and (8) M. teres major and M. dorsalis scapulae inset with common tendon on humerus. The concavo-convex frontoparietal suture, only in A. palustris, would be a reverted state in the other members of the clade ‘Allodaposuchia’ (Blanco et al., 2014).

The inclusion of Casa Fabà Eusuchian within the genus Allodaposuchus is well supported by the phylogenetic analyses (Fig. 8) and qualitative data (see above). Allodaposuchus differs from all other Eusuchians by the exclusive combination of the following synapomorphies (Blanco et al., 2014): margin of the orbits upturned; quadrate and squamosal not in contact on the external surface of the skull, posteriorly to the external auditory meatus; caudal margin of otic aperture not defined and gradually merging into the exoccipital; dermal bones roof overhang rim of supratemporal fenestra; cranioquadrate passage or canalis quadratosquamosoexoccipitalis laterally open and represented by a sulcus (broader than in Hylaeochampsa vectiana), with the exoccipital between the squamosal and the quadrate posterior to otic aperture. The ventral process of the exoccipital is not involved in the basioccipital tubera; the quadrate foramen aëreum on the dorsal surface. When preserved, all of the characters found in the specimen of Casa Fabà are in agreement with the synapomorphies of Allodaposuchus.

Allodaposuchus hulki, A. precedens and A. subjuniperus share a linear frontoparietal suture [C. 151 (1)], the presence of a shallow fossa or shelf at anteromedial corner of supratemporal fenestra [C. 153] (posteriorly reverted in Allodaposuchus hulki [C. 153 (1)]), and the exoccipital with very prominent boss on the paraoccipital process [C. 174 (0)].

Allodaposuchus hulki and A. precedens share the incisive foramen abutting the premaxillary tooth row [C. 89 (1)], external naris opened in aterodorsal direction [C. 81 (0)], and a very large medial jugal foramen [C. 102 (1)]. Such foramen is larger in A. hulki than in A. precedens, and consequently than in any other Allodaposuchus.

Allodaposuchus hulki shows some autapomorphies compared to other members of the clade ‘Allodaposuchia’: the quadratojugal that does not extend along the infratemporal fenestra [C. 145 (1)], the absence of fossa or shelf at anteromedial corner of supratemporal fenestra [C. 153 (1)], and teeth with characteristic morphology. In contrast, A. precedens shows dermal bones of the skull roof not overhanging supratemporal fenestra rim [C. 152 (0)], and capitate processes of laterosphenoid that are laterally oriented [C. 166 (0)]. In addition to the characters provided in the phylogenetic matrix, several qualitative morphological characters could be added to the phylogenetic results (see above). All of these differences justify the assignment of the specimen from Casa Fabà to a different species within Allodaposuchus.

Cranial pneumaticity and paratympanic recesses

In recent times, the virtual reconstruction of cranial cavities of extant and extinct archosaurs has provided an enormous advance in knowledge about the configuration and evolution of the brain regions and the surrounding bony recesses. Most of these studies focus on the endocast morphology as well as the inner ear disposition and its paleobiological implications (e.g., Witmer & Ridgely, 2009; Kley et al., 2010; Fernández et al., 2011; Bona, Degrange & Fernández, 2013, and references therein).

Despite lacking most of its ventral and posterior-most parts, the general shape of the cranial endocast of Allodaposuchus hulki (Fig. 7) is similar to those of extant crocodilians (e.g., Gavialis gangeticus Gmelin, 1789, Crocodylus johnstoni, Alligator mississipiensis; see Wharton, 2000; Witmer et al., 2008; Witmer & Ridgely, 2009; George & Holliday, 2013, respectively). Furthermore, the overall configuration of the cranial endocast of the specimen resembles that of many other crocodylomorphs such as notosuchians (e.g., Anatosuchus Sereno et al., 2003, Araripesuchus Price, 1955, and Simosuchus Buckley et al., 2000; see Sereno & Larsson, 2009; Kley et al., 2010, respectively) and metriorhynchids (Fernández et al., 2011). The curvilinear dorsal counter of the endocast exhibited by Allodaposuchus hulki is more similar to those observed in Neosuchia (Witmer & Ridgely, 2008; George & Holliday, 2013; Fig. 7A) than the characteristic spade-shape outline showed by notosuchians (Sereno & Larsson, 2009; Kley et al., 2010). In sagittal view, the shape of the cranial cavity indicates that most of the braincase elements were arranged in a planar configuration (Figs. 7D and 7E), in contrast to the sigmoidal organisation of most of the extant crocodylians Witmer et al., 2008; Witmer & Ridgely, 2009; George & Holliday, 2013. Another significant feature of the braincase of Allodaposuchus hulki is that in dorsal view, the cerebrum exhibits a rhomboid shape, which is more elongated rostrally than extant crocodilians, and shows a gentle transition to the olfactory track (Figs. 7A and 7B).

In comparison to works analysing braincase morphology, there are few studies focusing on the system of pneumatic cavities surrounding the main endocranial body (Witmer et al., 2008; Witmer & Ridgely, 2009; Witmer & Ridgely, 2010; Bona, Degrange & Fernández, 2013). In extant crocodilians, the paratympanic system is divided in the three main parts: (1) two caudal tympanic recesses connected by (2) the inner tympanic recess, which also links (3) the dorsal tympanic recesses located at each side of the endocranial cast (Witmer et al., 2008). The same configuration is observed in the Miocene caimanine Mourasuchus nativus Gasparini, 1985 (Bona, Degrange & Fernández, 2013). Dufeau & Witmer (2007) noted several ontogenetic changes in the tympanic cavities along the life history of Alligator mississippiensis. The authors stated that these changes could also be phylogenetically tracked within the crurotarsia linage, as basal taxa show the young alligator condition, whereas more crownward taxa resemble the adult one. In this regard, A. hulki shows endocranial features of both juvenile (i.e., large quadrate sinuses) and adult alligators (i.e., well-developed dorsal tympanic recesses).

In addition, Allodaposuchus hulki exhibits some important differences in the tympanic system configuration observed in extant adult crocodilians. First, it shows a well-developed pneumatic cavity connecting anteriorly both dorsal tympanic recesses. This sinus, herein referred as anterodorsal tympanic recess, covers part of the sphenoparietal dural venous sinus but leaving a circular opening at the level of the occipital dural venous sinus (Fig. 7A). In some ways, it resembles the frontal recess observed in the Struthio camelus Linnaeus (Witmer & Ridgely, 2009), but not so developed in Allodaposuchus. Although variations in the paratympanic system are reported in more derived archosaurs (i.e., some non-avian theropods exhibit supraoccipital pneumatic sinus connected to the tympanic pneumaticity, and birds have enlarged both dorsal and caudal tympanic recesses; Witmer & Ridgely, 2009; Witmer & Ridgely, 2010), nothing like the anterodorsal tympanic recess of A. hulki has been recognised so far. In addition, another distinctive feature of the tympanic system of the new species is the caudolateral expansion of the caudal tympanic recesses; which excavate a large cavity within the exooccipital bones. Such caudal tympanic sinus configuration is not present in any extant crocodilomorph, neither has it been reported previously from any extinct one, but it resembles that of large non-avian theropods (Witmer et al., 2008; Witmer & Ridgely, 2009; Witmer & Ridgely, 2010).

The external ear of the genus Allodaposuchus is distinguished by a broad cranioquadrate passage, a feature shared with the basal Eusuchian Hylaeochampsa (Buscalioni et al., 2001; Delfino et al., 2008; Blanco et al., 2014; Puértolas-Pascual, Canudo & Moreno-Azanza, 2014). Although its external morphology has been described in detail by several authors, here we provide the first tridimensional reconstruction of this cranial cavity and its relationship with the tympanic sinus and the braincase (Fig. 7). The cranioquadrate passage of Allodaposuchus hulki opens to a well-developed cavum tympanicum proprium that is lateromedially directed (Fig. 7A), which connects to the tympanic complex at the level of the dorsal tympanic recesses. In Crocodylus johnstoni, the cavum tympanicum proprium is more medioventrally directed (Witmer et al., 2008), a condition that is also noted in the caimaninid Mourasuchus (Bona, Degrange & Fernández, 2013). It is worth commenting that the cavum tympanicum proprium of Allodaposuchus hulki is ventrally connected to a large quadratic sinus (Fig. 7C).

In fact, the large size of the quadratic sinus is another highlighted feature of Allodaposuchus. In Crocodylus johnstoni, the quadratic sinus is also located above the cavum tympanicum proprium but it is smaller than that of the A. hulki and extends posterodorsally to the pharingotympanic recess (Witmer et al., 2008). A long and thin siphonial tube (see Witmer et al., 2008: Fig. 6.6.A-B) runs along the quadrate connecting the quadratic sinus and the siphonium. The later connects with the articular recess of the mandible. The siphonial tube is not observed in A. hulki, but the place where it was supposed to be is partially occupied by the enlarged quadratic sinus. Thus, we hypothesise that it could be relatively short, extending from the caudal end of the quadratic sinus and the foramen aëreum placed near the posterior edge of the quadratic hemicondyle.

The unique cranial pneumaticity configuration observed in A. hulki, especially with regard to the paratympanic system (e.g., an anterodorsal tympanic recess and enlarged caudal tympanic recesses) and the enlargement quadratic sinus, suggests some degree of otic specialisation. Although being just a hypothesis, the paratympanic configuration of A. hulki could result in a more efficient pressure difference receiver mechanism, which is related to directional hearing (Bierman et al., 2014). We hope that future studies could shed light on this question.

Forelimb myology and functional morphology

Three types of limb posture are traditionally identified in quadrupedal tetrapods during terrestrial locomotion: sprawling, semi-erected, and erected. Extant crocodilians locomotion ranges from sprawling, in which the limbs are positioned laterally, to a semi-erected high walk, in which limbs are strongly adducted (Briknam, 1980; Parish, 1986; Parish, 1987; Gatesy, 1991; Allen et al., 2014), whereas some extinct crocodylomorphs, such as notosuchians, could exhibit the erect posture (Sertch & Groenke, 2011; Chamero, Buscalioni & Marugán-Lobón, 2013; and references therein). However, most of these postures and locomotion inferences are based on anatomical features of the pelvic girdle and hind limbs rather than the pectoral girdle and forelimbs. In addition, the reconstruction of appendicular musculature is also regarded as being increasingly important in understanding locomotion behaviour in fossil vertebrates. Thus, recognising the morphological features in anterior limbs that characterise each type of limb posture is a clue to infer not only locomotion but also the lifestyle of Allodaposuchus hulki.

From a morphological point of view, the scapula of A. hulki exhibits a robust appearance. It is primarily characterised by having a well-developed anterior process with a wide deltoid crest, a marked scapular buttress, and scapular blade margins flared dorsally. Although no quantitative analyses can be conducted because of the fragmentary nature of the element, the combination of those features is consistent with the general scapular configuration of extant alligatorids and gavialids according to Brochu (1997) and Chamero, Buscalioni & Marugán-Lobón (2013), whereas crocodiles tend to show more slender scapulae with narrow blades. The presence of a prominent scapular buttress has been considered characteristic of upright posture in several terrestrial crocodilomorph taxa, whereas the absence of this treat is characteristic of primarily aquatic ones (Sertch & Groenke, 2011; Chamero, Buscalioni & Marugán-Lobón, 2013). Although this feature is present in Allodaposuchus hulki, it is not as developed as in terrestrial mesoeucrocodylian (e.g., Simosuchus or Araripesuchus). Thus, it may indicate some degree of upright posture, but not fully erect, or additional bracing of the forelimb. The angular morphology of the glenoid fossa suggests a moderate range of rotation of the humeral head within the glenoid cavity, which also agrees with a non-fully erect posture. Accordingly, sprawling or the semi-erected postures are the most likely terrestrial locomotion model to be inferred in A. hulki.

Another important trait of the scapula of A. hulki is the presence of several rough surfaces related to muscular attachment. Especially noteworthy are those located in the anterior process of the scapula, such as M. supracoracoideus intemedius, M. supracoracoideus longus, and M. coracobrachialis brevis dorsalis, which occupy relatively more of the surface than in extant crocodilomorph (Figs. 5A–5C; see Meers, 2003). These muscles are primarily involved in stabilising the shoulder joint, but they are also powerful protractors and adductors of the humerus, and may assist in extension of the forelimb. The M. deltoideus clavicularis, another powerful protractor of the humerus, is not specially developed in A. hulki, but muscle insertions are strongly marked. As a result, the morphology and myology configuration of the scapula of A. hulki seems to point at a powerful shoulder, with strong protractor/adduction capacities capable of supporting a robust body, keeping it off the ground.

Like the scapula, the humerus has an overall robust aspect, and it could be relatively short (Figs. 5D–5G). Because it lacks both epiphyseal ends, few assessments in regards to the torsion of the shaft or development of the articular parts can be performed. Furthermore, the preserved humerus does not exhibit any distinctive feature in its diaphysis if compared with extant crocodilians. Apart of the wide rough areas related to the M. scapulohumeralis caudalis, and the triceps brachii muscle complex, the humerus is a nearly smooth (Figs. 5D–5G). The M. scapulohumeralis caudalis assists in the elevation of the humerus and its stabilisation within the glenohumeral joint, but also plays an important role in protraction of the humerus. The main function of triceps brachii complex is to assist in the flexion of the brachium on the shoulder while extending the antebrachium on the brachium, thus supporting the body off the ground against gravity (Meers, 2003). These muscular features are in line with previous ideas suggesting that A. hulki could have robust forelimbs capable of performing powerful protractor movements during terrestrial locomotion.

The ulna is the most distinctive element of the forelimb of Allodaposuchus hulki. It is featured by an expanded proximal epiphysis with prominent processes, a shaft compressed mediolaterally with wide grooves located in both medial and lateral sites, and a marked twist of the distal end of the shaft. Overall, the ulna of A. hulki resembles those of extant crocodile taxa, but its expanded proximal epiphysis and twisted distal part resemble that of Simosuchus and Sebecus (Sertch & Groenke, 2011). As previously stated, the most prominent rugosity of the ulna is located in the olecranon process, corresponding to the insertion of M. triceps brachii (Figs. 5H and 5I; Meers, 2003; Allen et al., 2014). Less marked are those areas related with the insertion of M. flexor ulnaris and M. pronator quadratus, although they occupy relatively more surface than in current crocodile taxa (see Meers, 2003). This would mean that A. hulki could exhibit powerful muscles related to the flexion and pronation of the antebrachium. The crest placed at the origin of the M. flexor digitorium longus also suggests that a complex mechanism is involved in flexion of the wrist. Although characteristics of the appendicular skeleton suggest that A. hulki was not suited to a fully erect posture, several features indicate a powerful forelimb capable of performing sprawling and semi-erect postures. Furthermore, the robust configuration of the forelimbs seems to be consistent with a terrestrial lifestyle, or semi-terrestrial, rather than a semi-aquatic one.

The semi-terrestrial lifestyle hypothesis for Allodaposuchus hulki is also supported by the high degree of pneumaticity observed in its skull. Although large cranial cavities of A. hulki, such as the caudal tympanic recesses and quadrate sinuses, seem to be primarily related to a specialised otic system, they could also play an important role in lightening the weight of the skull like in large non-avian theropods (see Witmer et al., 2008; Witmer & Ridgely, 2009; Witmer & Ridgely, 2010), or other terrestrial vertebrates.

Ecological implications

According to both cranial and postcranial features displayed in Allodaposuchus hulki, it could exhibit some kind of terrestrial or semi-terrestrial lifestyle rather than semi-aquatic ones. This interpretation is also supported by paleoenvironmental evidence.

Charophyte fructifications at the Casa Fabà outcrop were found in the grey claystones above the pedogenised channelised sandstone bed belonging to the lower part of the fluviatile ‘lower red unit’ (Riera et al., 2009). The charophyte assemblage is formed by extremely small gyrogonites of Microchara cristata Grambast, Microchara nana Vicente & Martín-Closas, Microchara punctata Feist & Colombo and Microchara aff. laevigata Grambast & Gutiérrez. Most of the samples show well-preserved gyrogonites, suggesting that they belong to an autochthonous assemblage. Charophytes were also found along with gastropod shells and operculi, fragmentary vertebrate remains and slightly eroded eggshells and planktonic foraminifera.

Assemblages formed exclusively by species bearing small gyrogonites (e.g., M. nana) have been related to turbid and warm ephemeral ponds usually found in terrigenic floodplains (Vicente et al., 2015). Despite also being common in lacustrine and palustrine environments, the absence of typically lacustrine species in the assemblages suggests that this highly fluctuant and stressed continental environment favours the thriving of these adapted species bearing small gyrogonites.

In addition, no channel or lake deposits have been found near the Casa Fabà site, or at least no closer than 2.5 km away. These evidences, along with anatomical characteristics, may suggest that Allodaposuchus hulki could perform relatively large incursions on the earth, moving from place to place, only stopping in ephemeral water bodies looking for food or other resources.

Supplemental Information

Supplemental Information S1 Codificatio of Allodaposuchus hulki

Click here for additional data file.

Supplemental Information S2 Data matrix used in the phylogenetic analysis

Click here for additional data file.

We thank the staff of Museu de Ciències Naturals de Barcelona for the temporary loan of extant specimens for comparison, Hospital Universitari Mútua de Terrassa for the CT-scanning, the Preparation Division of the ICP for the preparation of the fossils specimens, Sergio Llácer for the image processing and AM Bravo (Museo Geológico de Madrid, Madrid) and R Gaete (Museu de la Conca Dellà, Spain) for permission to study the fossil material. We also acknowledge Víctor Fondevilla and Eduardo Puértolas for their useful comments.

Institutional Abbreviations

MCD Museu de la Conca Dellà, Lleida, Spain.

MZB Museu Zoològic de Barcelona, Barcelona, Spain.

Additional Information and Declarations

Competing Interests

Author Contributions

New Species Registration

The authors declare there are no competing interests.

Alejandro Blanco and Albert G. Sellés conceived and designed the experiments, performed the experiments, analyzed the data, wrote the paper, prepared figures and/or tables, reviewed drafts of the paper.

Josep Fortuny performed the experiments, analyzed the data, wrote the paper, prepared figures and/or tables, reviewed drafts of the paper.

Alba Vicente analyzed the data, wrote the paper.

Àngel H. Luján prepared figures and/or tables, reviewed drafts of the paper.

Jordi Alexis García-Marçà wrote the paper.

The following information was supplied regarding the registration of a newly described species:

Allodaposuchus hulki urn:lsid:zoobank.org:act:267AADFA-AD84-45F4-B195-D08E174559CC.

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
