# Peer review of "A new species of Allodaposuchus (Eusuchia, Crocodylia) from the Maastrichtian (Late Cretaceous) of Spain: phylogenetic and paleobiological implications"

_PeerJ, doi:10.7717/peerj.1171_

## Round 0.1 · original submission · Major Revisions

The two reviewers agree that your manuscript is interesting, but they also agree that several parts of it should be modified and improved. The major issue concerns your diagnosis of Allodaposuchus hulki and the limitation of the comparisons given the poor preservation of your material. The two reviewers actually question whether you truly have a new species in hand. At the very least, you must provide a differential diagnosis that clearly tells in which features A. hulki is different from each other species of Allodaposuchus. You must also improve your comparative section, paying special attention to the possible effects of individual variations and poor preservation. If you are still confident that you can define a new species based on this material, you need to address these issues. Alternatively, you may want to consider Massimo Delfino’s suggestion to not erect a new species based on this material and focus on the other morphological aspects of your study.

The two reviewers have done an excellent job and provided very detailed reports. I strongly urge you to revise your manuscript following their suggestions and comments. Your manuscript should also be checked by a native English speaker. Please, do not consider this last comment lightly. There are numerous issues with English in your manuscript.

·

Basic reporting

No Comment

Experimental design

no comments

Validity of the findings

The paper describes new material of Allodaposuchus from Spain. This taxon is particularly important to resolve the phylogenetic relationships of Crocodylia. Many characters are provided to justify the erection of the new species, but as the material is poor (as for many species of the genus), the comparisons are difficult. As often when the material is fragmentary, the authors try to find diagnostic characters, and often, in my opinion, use poor characters (intraspecific variability, weak differences…) (see below). So, I am not convinced by this new species.
At least, the presentation of the characters should be revised, and a comparative diagnosis would be clearer when the character differs from only one other Allodaposuchus species (differs with … in having…).
The phylogenetic analysis is well conducted, but I strongly disagree with the phylogenetic results obtained by the authors, and with some of the coding, but it is beyond the scope of this review, and that will be discussed in a paper in preparation. Nevertheless, the results should be more extensively discussed (see below), and the matrix in a nexus format should be provided.

I think some grammatical mistakes are present in the text, but I am not a native English speaker. So, the English should be reviewed.

Additional comments

Detailed comments

L23: Crocodilomorphs: Crocodylomorphes

L149: Diagnosis:

Allodaposuchus hulki shows the following autapomorphies:
Quadratojugal does not extend along the infratemporal fenestra. : known only in A. subjuniperus, unknown in other species. It is not possible to know if it is an autapomorphy.

Spine of quadratojugal significantly reduced: known only in A. subjuniperus. Is its preservation allow a clear observation in present specimen?

Absence of fossa at anteromedial corner of the supratemporal fenestra.
L448: Additionally, A. hulki and A. palustris are the two species that do not show a fossa around the supratemporal fenestrae. As it is present only in A. subjuniperus and precedens, its absence is a plesiomorphy. Moreover, its presence or absence cannot be observed in the present specimen, as the fossa is within the supratemporal fenestra, this one being obscured by sediments.

No ridge surrounds the foramen aërum.
L455: in A. hulki no ridge surrounds the foramen aërum, unlike the other three species of Allodaposuchus: No ridge in A. precedens. And not sure that the difference was such significant to be considered as anything else than individual variation.

No elevation rounds the rim of external naris: also absent in A. precedens and it is weak in A. subjuniperus (slightly elevated). So if this character is diagnostic (I doubt…) it is its presence in A. subjuniperus, not its absence in other Allodaposuchus.

Incisive foramen abuts first and second premaxillary teeth. The distance 1st tooth-FI is equal to the 1st tooth diameter in A. subjuniparus and A. precedens, and slightly lesser in the new specimen. The difference is weak, and it strongly depends on the angle of the photo in A. subjuniperus, and A. precedens.

Teeth bear smooth enamel, low-developed mesial and porterior carinae, and absence of longitudinal grooves in lingual side. Related to intraspecific variability, and unsignificant differences with A. subjuniperus (see below). The lingual groove is present only in Premaxillary teeth in A. precedens. So, the tooth ornamentation varies along the tooth row, and the difference (if there is) is only with A. precedens…..

Exclusive combination of the following synapomorphies: External naris opens in anterodorsal direction: as in all Allodaposuchus, except in A. subjuniperus…. It should a A. subjuniperus autapomorphy, but must not be included in the present diagnosis.

Premaxilla is wider than long: as posterior process of the premaxilla is not preserved this cannot be evaluated. But I think that the authors mean that the premaxilla is wide laterally to the external nares. This is mainly du to the size of the external nare, large in A. precedens (Delfino, 2008 and Martin, 2010), but it is comparable here to this is observed in A. subjuniperus, and Acynodon…. So, this is more probably a plesiomorphy. If the authors mean that the premaxilla is wider than the length from tip of snout to lateral premaxillary-maxillary suture, this is also the case in all other Allodaposuchus. Moreover, the posterior margin of the premaxilla seems to be strongly damaged. Is the premaxillo-maxillo suture preserved ? I am not sure on the figures.

Four premaxillary alveoli, being the third the largest. As in A. subjuniperus.

Premaxillary-maxillary suture does not reach the posterior margin of the incisive foramen. As in all Allodaposuchus ! Even if the distance between the suture and posterior margin of the foramen insicivum is longer in A. subjuniperus and present form compared to A. precedens, it is also large in Acynodon. This character (if you don’t retain the possible intraspecific variability), could be a A. subjuniperus autapomorphy.
L430: In palatal view, the incisive foramen is located more anteriorly than A. precedens and A. subjuniperus. No, the FI is not more anterior than in A. precedens .

Wide and short orbits, without interorbital ridge in the frontal. How can you evaluate the size of the orbits, when they are not preserved ?? To evaluate the orbital width, its lateral margin is required! Moreover, comparing the authors reconstruction with A. subjuniperus and A. precedens….. the difference is weak or absent. Not sure that it is possible to exclude the presence of a frontal crest. The anterior portion is absent herein, and it cannot be excluded that it was present anterior to the preserved portion (see A. subjuniperus).

Quadrate bears two crests in ventral surface for muscle attachment. As in many eusuchians…. Its absence should be a A. palustris autapomorphy.

Capitate process of the laterosphenoid is anteroposteriorly oriented.
L463: In ventral view, the capitate process of the laterosphenoid is anteroposteriorly oriented in A. hulki and A. subjuniperus, but is laterally oriented in A. precedens. This character is not clear. See Buscalioni et al. (2001) Fig. 2, where the capitate process is anteroporteriorly oriented….

Aside of the previous characters, A. hulki has the following ambiguous autapomorphies: Anterolateral, anteromedial and olecranon processes of the ulna well developed. Ulnar shaft lateromedially compressed with lateral and medial grooves. Distal condyles of the ulna turned lateroposteriorly, causing a lateral crest in the shaft. We prefer coding all these autapomorphies as ambiguous, due to the absence of postcranial remains in other species of Allodaposuchus. New discoveries may reveal if they are autapomorphies of the genus.
As they cannot be compared with other Allodaposuchus (as many other Eusuchians), these characters should not be included in the diagnosis.

L180: quatdrates: quadrates

L181: quatdratojugal: quadratojugal

L217: In lingual view, an uncommon large medial jugal foramen: equivalent to the jugal foramen observed and coded as large in A. precedens (cf Brochu, 2011; char. 102)?

L224: Quadratojugal spine is nearly absent and low in position: preservational artifact ?

L234: Both articular hemicondyles are similar in size, although the medial hemicondyle is slightly smaller and ventrally deflected: in Fig. 2C-D, the quadrate condyle is figured as damaged…. How their shapes can be evaluated ?

L276: lateral margin of the supraorbital fenestra: supratemporal fenestra ?

L330: In the same side, but in a most ventrally position : In the same side, but in a more ventral position ?

L331: and immediately superior to supraglenoid: and immediately dorsal to supraglenoid ?

L346: the humeral head seem to have the same high: the humeral head seem to have the same height.

L351: several rugous areas around itself : not clear…… what does that mean?

L370: In proximal view is triangular-shaped: In proximal view it is triangular-shaped

L405: decrease in high respect the centrum: decrease in height respect the centrum… ? not clear.

L430: In A. hulki the anterior rim of incisive foramen is located between the first and second alveoli, whereas in A. precedens and A. subjuniperus, reaches the third premaxillary alveolus.
It depends on the orientation of the fragment. If the ventral view is oriented as the dorsal view, the anterior margin is located at the same level as the 3rd tooth (see below).

L435: Moreover, A. hulki shows a pattern of oclusal pits different to A. precedens and A. subjuniperus. How ? Does not seem so different…






L446: The frontoparietal suture of A. hulki is nearly linear, like A. subjuniperus and A. precedens. Allodaposuchus palustris is the only ‘allodaposuchian’ that shows a concavo-convex frontoparietal suture. The suture is not really concavoconvex in A. palustris (compare with Crocodylus), and no more than the specimen described herein (see below).


L450: Like A. precedens, both articular hemicondyles of the quadrate of A. hulki are similar in size, although the medial hemicondyle is slightly smaller and ventrally deflected. In turn, A. subjuniperus shows a ventral expansion in the medial hemicondyle. in Fig. 2C-D, the quadrate condyle is figured as damaged…. How their shape can be evaluated ? The medial hemicondyle does not seem to differ between A. subjuniperus and described form.


L452: The foramen aërum of A. hulki is small, like A. precedens and A. subjuniperus, but large in A. palustris. Not significantly larger in A. palustris (evident compared with A. subjuniperus below). More probably individual variations.



L465: Unlike A. precedens and A. subjuniperus, quadratojugal does not extend to superior angle of infratemporal fenestra. Due to poor preservation, the participation of the quadratojugal is not larger in A. subjuniperus than in the present form, and in A. precedens is not so longer.




L469: The dentition of A. hulki also shows several characters that distinguish itself from the other species of the genus. In A. hulki, the enamel lacks ornamentation, both anterior and posterior carinae are poorly developed, and there are not longitudinal grooves in lingual side. In contrast, A. palustris and A. precedens show ornamented enamel with well-developed carinae, whereas A. subjuniperus and A. palustris bear well-marked longitudinal grooves in lingual side. First, the size of the carinae can vary according to the position of the tooth in the tooth row (as the ornamentation of the enamel). The tooth enamel of A. subjuniperus is described as smooth. Moreover, the ornamentation is weak, smooth in A. palustris, and micrometric in the apex, and the difference with the specimen described herein is probably not significant (micrometric ornamentation also ?). Differs only with those of A. precedens (considering the differences between the premaxillary and maxillary teeth…).

Discussion
Phylogeny

L547: The most parsimonious hypothesis obtained in our analyses suggests that the clade ‘Allodaposuchia’ belongs to Crocodylia……….L567: would be a reverted state in the other members of the clade ‘Allodaposuchia’ (Blanco et al., 2014) : All this paragraph is nearly identical to paragraph from the discussion in Blanco et al., 2014…..! This paragraph is not clear ! It would be more useful to discus the character distributions in the results obtained by the authors, according to the different position of the Allodoposuchus in the trees (unresolved position of Allodaposuchus in the consensus tree, but what are their distributions in the various trees obtained??? Which consequences on the character distributions and history ??). I do not understand if the characters listed in the paragraph are those that support the presence of Allodaposuchus within Crocodylia, or those proposed by Brochu ? How these characters are distributed in your trees ??

Moreover, some of the characters cited are problematic:
3) frontoparietal suture concavo-convex: the suture is not concavoconvex in Allodaposuchus.
1) slender postorbital bar: The postorbital bar is huge in Allodaposuchus.

L573: supretmporal: supratemporal

L664: Crocodrylus: Crocodylus


S. Jouve

·

Basic reporting

See below

Experimental design

See below

Validity of the findings

See below

Additional comments

Cretaceous crocodylomorphans recently attracted very much the interest of researchers because it is in the Cretaceous that extant crocodilian clades arose. Among the Late Cretaceous European taxa, new Allodaposuchus remains were found during the last 20 years in several French, Spanish and Romanian localities and therefore the knowledge of this taxon significantly improved.
This manuscript deals with new Allodaposuchus remains coming from a Maastrichtian Spanish locality. The authors referred all the remains to the fragmentary skeleton of a single specimen represented by a poorly preserved skull and lower jaw (poorly preserved both in terms of amount of available skeletal elements and condition of the bone surface) and a few postcranial elements.
The authors developed their research by describing the material, identifying it as a new species, evaluating its phylogenetic relationships, describing the inner structures of the skull (thanks to CT-scan data) and describing the forelimb miology and functional morphology.
The weakest point of the manuscript are the diagnosis of the new species and the discussion of its comparative morphology. Conversely, the description inner structures of the skull and the sections about the forelimb miology and functional morphology represent a significant advance in the knowledge of Allodaposuchus because the manuscript present the first information on these topics.
Besides the fact that the diagnosis is not differential and therefore does not allow the reader to immediately understand the differences between this “species” and the species already identified, it seems very likely (I didn’t see the material) that all the characters described are just intraspecific variation or related to the preservation of the material.
In particular: the quadratojugal spine is subject to some sort of variation in extant crocodilians and seems to be very poorly preserved in the fossil remains here described; I would say that it is very difficult to evaluate the narial morphology on the basis of the fragmentary isolated premaxilla that is available; the ridge corresponding to the foramen aërum is a too variable character to be included in a specific diagnosis; the morphology of the teeth varies very much in the material associated to the type specimen of Allodaposuchus precedens (and the few available teeth preserved in the material here described are very badly preserved / or replacement teeth not fully formed).
The characters “absence of fossa at the anteromedial corner of the supratemporal fenestra” should be better clarified, because for sure there is a fossa (that is to say the hole defined by its rim, the fenestra) but maybe not a SHELF at the anteromedial corner. This character is not visible in Fig. 2 because of the matrix that still fills the fossa but the authors probably explored this morphology thanks to the CTscan. It would be important to see this character in detail in a figure. However, it must be considered that some variation (intraspecific and/or ontogenetic) in this structure is present in some still unpublished skulls of A. precedens from Souther France.
Before erecting a new species the authors should mandatorily study directly the type material of Allodaposuchus precedens (Budapest), the well preserved skull from Romania (Cluj-Napoca), as well the French and Spanish remains.
As for the phylogenetic analysis, it is clearly stated that some modifications were done to the dataset by Brochu (2011). It would be important to know if the topology you got by running the analysis WITHOUT your new material is congruent to that obtained by Brochu (2011) or if the new “unconventional” topology you obtained is the direct effect of the inclusion of the new material Worth noting is that according to the results of your analysis the “new species” clusters with A. precedens and therefore it could be well the same species.
Moreover, in fig. 2B the capitate processes are asymmetric and could be laterally directed and not anteroposteriorly oriented. I would suggest to double check this character and maybe run again the analysis with this change.

My suggestion is simply to not erect a new species on the basis of this fragmentary remains but focus the manuscript on the new information that the CT-scan and the analysis of the muscles can provide. This would make the manuscript much more robust and a significant contribution to the knowledge of the morphology, functional anatomy and palaeoecology of Allodaposuchus.

General note: the grammar of the manuscript has to be mandatorily reviewed by an English native speaker.

References:
- please not the pages of the cited papers are in some cases separated by a hyphen but in others by a n-dash.
- Minor issues are reported below.

Figures:
- The figures are generally excellent but the absence of symmetry in the lines corresponding to the sutures of Fig. 2 seem to indicate that the sutures are not actually visible on the fossil material. If so, it would be better to use dashed lines and not solid lines.
- Fig. 2: note that the abbreviation “A” is not present in the caption.
- Fig. 3. Note that it would be better use the singular and not the plural for “dal” and “ids” because each line indicates respectively only one alveolus and one septum.
- Fig. 4. “dacu” in the caption -> is actually “dac” in the figure; “dpcu” in the caption -> is actually “dpc” in the figure; “mgr” in the caption -> is actually “mg” in the figure. Moreover, I did not succeed in finding in the figure the following abbreviations that are present in the caption: “dc” and “gr”.
- Fig. 5: “ecrd-pu” in the caption -> is actually “ecrb-pu” in the figure. Moreover, I did not succeed in finding in the figure the following abbreviation that is present in the caption: “ld”.
- Fig. 6. I suggest to rotate of 180° the photos in dorsal view: C, G, K, O. (see all the previous literature on fossil vertebrates and the vertebrae are always figured with the posterior sector at the bottom of the figure). Moreover: channel –> canal.
- Figure 7. craniocuadrate -> cranioquadrate
- Figure 8. Just a very minor detail: note that the genus Baryphracta was recently synonymized with Diplocynodon.

Punctual issues to be considered are listed below:
Abstract:
- the usage of the term CROCODILES is not correct. This term refers to the members of the clade CROCODYLIDAE whereas here the authors would clearly like to mean CROCODILIANS [note that the term CROCODILES was used correctly at line 701];
- performer -> performed;
line 19. crocodiles -> crocodilians
line 32. taxon -> taxa
line 34-35. I suggest to write the full name of the genus when a species is mentioned for the first time in a section; moreover, since the paper has a taxonomic goal it would be important to cite the authors of the species at first mention
line 42. Allodaposuchus palustris -> A. palustris
line 46. treats -> traits
line 81. Were -> why plural?
line 81. it is not clear which is the subject of REPRESENTED (it seems that it is the area)
line 90. Please check all the refences and the authors of taxa because sometimes a comma separates the authors from the year and sometimes not.
line 98. This whole section (Descriptive anatomy) has to be checked. Maybe a title like “Anatomical nomenclature” would be more appropriate. Please note that the words “according to these studies” correspond to references included in a parenthesis.
Line 101: phylogenetical -> note that in line 105 you wrote phylogenetic
Line 110. Remove the parenthesis before Goloboff
Line 149. The Diagnosis is not differential. This should be mandatorily changed.
Line 151. foramen aërum -> foramen aëreum [see Iordansky, 1973]; moreover, why you use italics for this foramen but not for foramen magnum [both are in latin]
Line 167. I think that it would be much better to indicate here and not in the section MATERIAL the correspondence between numbers and remains
Line 174-177: this is no description but just a repetition of what written in the lines 83-89
Line 186. remove comma: located between
Line 189. There is no fossa around the supratemporal fenestrae -> there is no shelf at the anteromedial corner of the supratemporal fossa.
Line 196. Please rephrase the first sentence.
Line 200. lateral and rim [?]
Line 205. being completely -> being the latter completely
Line 206: better to avoid terms like UNFORTUNATE that betray what the authors are hoping for.
Line 207. Alveoli -> alveolus
Line 212. posterorbital -> postorbital
Line 217. Uncommon -> uncommonly
Line 240. Please note the term IMMEDIATELY refers to time and no to the position of a certain structure
Line 249. strongly concave medially -> not clear. Do you mean “at the center of the dorsal surface”?
Line 255-256. how that suture could not enter the fenestra caudally? [I suggest to delete the sentence]
Line 259-260. It that recess visibile in the figures?
Line 276 and 294: conforming -> please check this term
Line 295-296. in fig. 2B the capitate processes are asymmetric and could be laterally directed and not anteroposteriorly oriented. I would suggest to run again the analysis with this change.
Line 310. Please list clearly how many teeth are available and
Line 314. Soft -> weak [?]
Line 317: note that the order of this heading should be different from that of the skeletal elements that follow.
Line 330. Other -> another [?]
Line 346. High -> height [¿]
Line 347. Lost: do you mean because of the preservation? (lost could also be applied to evolutionary changes)
Line 350: the term BEFORE refers to time. Please use anatomical terms instead.
Line 395. Broken laterally -> not clear [the lateral –but which side?- area is broken off?]
Line 405. Respect the centrum -> respect to the centrum
Line 409-410. The same sentence was written at line 401
Line 415. Comprised -> compressed [¿]
Line 426. Oclusal -> occlusal
Line 426. Please provide details about the different location of the pits
Line 448. Fossa around -> no; see above
Line 479. Due to the former; please rephrase
Line 501-502. There is a lot of literature concerning the fact that the appendicular skeleton is conservative. I suggest to provide a broader spectrum of references.
Line 508. The first sentence should be supported by at least one reference.
Line 512. The term WHILE refer to time. Use WHEREAS instead (here and elsewhere)
Line 541. Genus -> genus
Line 543. Revised -> maybe better something like: confirmed
Line 585 and following lines. I’m not sure about the need to repeating here all the info that were already included in the diagnosis.
Line 602. Please note that the many extant crocodilians show only 4 teeth late in ontogeny (simply because the third can crowd out the second during growth. In order to express general consideration like the one reported here, it would be necessary to find a juvenile from the same locality.
Line 664. Crocodrylus -> Crocodylus
Line 698. Butter -> buttress
Line 840. Remove “a” after the date
Line 849. Not sure that some of the words of the title of the journal have to start with a lower case
Line 888. Lower case for “crocodilian”
Line 904. Simosuchus clarki is not in Italics.

---

## Round 0.2 · Minor Revisions

First, I would like to apologize for the delay, this is entirely my fault.

You have made significant improvements to this revised version. You have followed (or replied to) most suggestions made by the two initial reviewers on the first version of this manuscript. As you will see, Stéphane Jouve is still not convinced that your material represents a new species, but you have made your arguments clearer and it would be easier for future workers to confirm or reject your conclusions. Stéphane Jouve provided several additional comments on this revised version. I will leave you to judge whether you want to consider them or not. He did put a lot of work on it, and you might want to consider some of his new remarks.

I have also carefully read your manuscript and made several comments on the attached pdf. When revising your manuscript, please be aware of the following:

- modifications to the first two pages (title and abstract) must be entered online
- years must be provided for taxonomical authorships and relevant publications must be listed in the reference list
- see at the end of the attached pdf for my remarks regarding figure captions, and enter them online as well
- File S2 is not cited in the text. You must cite it somewhere.

·

Basic reporting

no comments

Experimental design

No comments

Validity of the findings

As I wrote it in my first review, the paper is good, but I still disagree with the erection of a new species. In my opinion, the material is too poor, and numerous characters could be related to intraspecific variability… (see below). In particular the differences with A. palustris are unclear (this species is also formed with fragmentary remains….). So, I am not convinced by this new species.
If it is not possible (in my opinion) to determine if the new material is a new species or not, this does not reduce the interest of the paper (on the contrary !), and I agree with Massimo, I recommend the authors to focus the paper on the description of the inner structures of the skull and the sections about the forelimb miology and functional morphology.
That’s why I recommend a major/minor revision (most of the work has been done by the authors).

Additional comments

See attached file for detailled comments

---

## Round 0.3 · accepted · Accept

I am satisfied with the changes you have made to the manuscript. Your paper is now accepted for publication.

Debate is part of science, but I regret that the tone was not always cordial during the review process for this manuscript. That being said, a lot of interesting discussion occurred during the review and I would like you to consider to make the review history public for the benefit of all.

Finally, I noted that file S2 was still not cited in the manuscript. You will be required to change this at proof stage.